# Relaxed Substrate Specificity in Qβ Replicase through Long-Term In Vitro Evolution

**DOI:** 10.3390/life12010032

**Published:** 2021-12-26

**Authors:** Kohtoh Yukawa, Ryo Mizuuchi, Norikazu Ichihashi

**Affiliations:** 1Department of Life Science, Graduate School of Arts and Science, The University of Tokyo, 3-8-1 Komaba, Meguro-ku, Tokyo 153-8902, Japan; kyukawa@g.ecc.u-tokyo.ac.jp; 2JST, PRESTO, Kawaguchi 332-0012, Japan; mizuuchi@bio.c.u-tokyo.ac.jp; 3Komaba Institute for Science, The University of Tokyo, 3-8-1 Komaba, Meguro, Tokyo 153-8902, Japan; 4Research Center for Complex Systems Biology, Universal Biology Institute, The University of Tokyo, 3-8-1 Komaba, Meguro, Tokyo 153-8902, Japan

**Keywords:** Qβ replicase, origin of DNA, experimental evolution, RNA world

## Abstract

A change from RNA- to DNA-based genetic systems is hypothesized as a major transition in the evolution of early life forms. One of the possible requirements for this transition is a change in the substrate specificity of the replication enzyme. It is largely unknown how such changes would have occurred during early evolutionary history. In this study, we present evidence that an RNA replication enzyme that has evolved in the absence of deoxyribonucleotide triphosphates (dNTPs) relaxes its substrate specificity and incorporates labeled dNTPs. This result implies that ancient replication enzymes, which probably evolved in the absence of dNTPs, could have incorporated dNTPs to synthesize DNA soon after dNTPs became available. The transition from RNA to DNA, therefore, might have been easier than previously thought.

## 1. Introduction

One of the major evolutionary events in primordial life would have been the change in the genetic material from RNA to DNA. It is hypothesized that RNA played the role of the genetic molecule in the ancient RNA and RNA–protein worlds, and that it was replaced by DNA in early evolution [1]. To date, several researchers have proposed possible scenarios for this transition [1,2,3,4,5,6,7,8,9]. Phylogenetic analysis of polymerases between three domains of life support the notion that DNA polymerases (DNAPs) are derived from an ancestral RNA polymerase (RNAP) [10]. However, it is still largely unknown how such a transition could have occurred.

One of the possible prerequisites for this transition is the change in the substrate specificity of an RNAP from ribonucleotide triphosphate (NTP) to deoxyribonucleotide triphosphate (dNTP). Standard RNAPs have strict substrate specificity; RNAPs selectively use NTPs even in the presence of a high concentration of dNTPs in the cell [11] because the selection of correct nucleotides is essential for their function. However, the situation might have been different in primordial cells that used RNA as the genetic material, and where dNTPs would not have existed. RNAPs that had evolved and worked in such an environment might not have had strict substrate specificity and therefore could have used dNTPs if available. Such loose substrate specificity might have been a common feature of ancient enzymes [12]. The RNAPs with low substrate specificity could have immediately started the synthesis of DNA once dNTPs appeared and thus would have allowed for an easier transition from RNA to DNA as the genetic material. Although ancient RNAPs remain unknown, we can obtain RNAPs in the absence of dNTPs, similar to those in the ancient situation, using an evolutionary experiment. Here, we hypothesized that RNAPs that evolved in the absence of dNTPs might be able to utilize dNTPs if they are available. To test this hypothesis, it is necessary to examine the substrate specificity of polymerases that have evolved in an unnatural dNTP-free environment.

Evolutionary experiments with replicating molecules have been conducted for decades. The first example is the evolution of the Qβ RNA by Spiegelman’s group, in which the genomic RNA of bacteriophage Qβ was continuously replicated by the RNA-dependent RNA polymerase protein (Qβ replicase) of the bacteriophage in a test tube and through a serial dilution cycle [13]. Mutations were randomly introduced into RNA due to replication errors, and mutant RNAs that replicated faster dominated the population (i.e., evolved). Recently, we developed the system by introducing a reconstituted translation system of *Escherichia coli* [14,15], by which the Qβ replicase is translated from a genomic RNA to replicate the RNA [16]. In previous studies, we encapsulated this reaction system in microcompartments and conducted a series of long-term evolutionary experiments. The reaction mixture did not contain dNTPs, and thus a series of mutant RNAPs that emerged during this evolutionary experiment would be ideal candidates for examining the above hypothesis. In this short communication, we examined the incorporation of labeled ribo- and deoxyribonucleotides by the original and three mutant RNAPs that appeared during the course of the evolutionary experiment.

## 2. Results

In our previous evolutionary experiments, we first performed 96 rounds of serial dilution cycles while repressing the amplification of parasitic RNAs (parasite-absent condition), i.e., small RNAs that lost the replicase gene and thus replicated relying on the replicase translated from other RNAs [17]. Next, we performed an additional 115 rounds of serial dilution cycles in the presence of parasitic RNAs (parasite-present condition) [18]. We changed the replication conditions from parasite-absent to parasite-present conditions to accelerate evolution by inducing evolutionary arms races between host and parasitic RNAs. It should be notable that although we did not intentionally introduce mutations, a sufficient number of mutations were introduced by replication errors. In this study, we used the following four RNAs in the course of these two serial dilution experiments: the original RNA (Original) before the first serial dilution experiment, the dominant RNA after 96 rounds of the first experiment (RNA1), and the most frequent RNAs in the population after 99 (RNA2) and 115 rounds (RNA3) of the additional serial dilution experiment with parasitic RNAs (Figure 1a). Nonsynonymous mutations in the Qβ replicase are listed in Appendix A (all mutations observed in RNAs are listed in Appendix A).

First, we compared the replication abilities of these RNAs. Each RNA (10 nM) was incubated in the reconstituted translation system of *E. coli* at 37 °C for 2.5 h. In this reaction, the RNAPs were translated from the RNA and they in turn replicated the RNA (Figure 1b). RNA concentration was then measured by quantitative PCR after reverse transcription (RT-qPCR) (Figure 1c). The concentration of the original RNA was found to be close to 10 nM, indicating that the original RNA barely replicated, while RNA1 was significantly amplified, which was consistent with a previous report [17]. The replication of RNA2 and 3 further increased as the number of rounds increased.

We next established a sensitive detection method for dNTP incorporation as dNTP incorporation by RNAP was expected to be very low. The procedure is illustrated schematically in Figure 2a. We incubated each RNA (10 nM) in the reconstituted translation reaction solution containing 1.1 mM ethynyl-labelled deoxyribonucleotide, 5-ethynyl-2′-deoxyuridine 5′-triphosphate (EdUTP), and 0.11 mM uridine-5′-triphosphate (UTP). Here, we used a deoxyuridine analog, not a natural deoxyribonucleotide, to evaluate the effect of the lack of the 2′-hydroxyl group. Translation of RNAP and RNA replication by RNAP occurs during the incubation period. If EdUTP is utilized by RNAP, the replicated RNA should contain ethynyl modification. Biotin was then covalently attached to the ethynyl group through a click reaction (Appendix A) [19]. Biotinylated RNA was recovered using avidin-coated magnetic beads. The RNA concentrations before the recovery step (i.e., the concentrations of replicated RNAs) roughly increased with the increasing number of rounds, although RNA2 exhibited a slightly lower concentration than RNA1 (Figure 2b), similar to the results obtained without using EdUTP (Figure 1c). The EdUTP incorporation ratio, the ratio of RNA concentrations after the recovery step to those before the recovery step, increased as the number of serial dilution rounds increased (Figure 2c), indicating that the RNAPs that experienced longer in vitro evolution tended to utilize ethynyl-labeled deoxyribonucleotides more efficiently.

To understand whether the above-mentioned trend is specific to labeled deoxyribonucleotides or is also observed for labeled ribonucleotides, we next measured the incorporation of 5-ethynyl-2′-uridine 5′-triphosphate (EUTP) into RNA. We used the cell-free translation reaction solution containing 1.1 mM EUTP and 0.11 mM UTP and performed the same experimental procedure. The pattern of the RNA concentrations before the recovery step was similar to the results obtained with EdUTP, but the absolute values were much lower (Figure 2d), probably due to the inhibition of replication caused by the more frequent incorporation of the unnatural nucleotide, EUTP. Consistently, the incorporation ratio, the ratio of RNA concentrations after the recovery step to those before the recovery step, was significantly higher than that of EdUTP (Figure 2e); the ratio increased as the number of serial dilution rounds increased, similar to the result obtained for EdUTP, suggesting that the RNAPs tended to utilize not only dNTP but also other non-canonical nucleotides as the in vitro evolution proceeded. We also attempted to measure the incorporation of larger labeling groups, namely Cyanine5, Cyanine3, and biotin; however, they were undetectable (data not shown), probably due to a larger modification size and/or low detection sensitivity compared to the ethynyl-labeling method.

There were 13, 18, and 17 nonsynonymous mutations fixed in the Qβ replicases encoded by RNA1, 2, and 3, respectively (Appendix A). These mutation sites are shown in Figure 3 in red. Qβ replicase has tunnels that allow NTPs to enter and be polymerized (indicated as “NTP tunnel”) [20,21]. The mutation sites were distributed throughout the structure. Some of the mutations in RNA1–3 (I283M, C366R, G124C, S351P, and V91A) were located near the catalytic site. To examine the effects of these mutations, we generated mutants that reverted one of the mutated residues back to the wild type and measured the incorporation of EUTP. We found no significant decrease in the incorporation of EUTP compared to the RNA before reversing the mutations (Appendix A), indicating that each of these mutations alone had negligible effects on substrate specificity. This result suggested that other mutations or a combination of multiple mutations alter the substrate specificity.

## 3. Discussion

We demonstrated that RNAPs that experienced evolution in the absence of dNTPs incorporated noncanonical substrates, namely, ethynyl-labeled dUTP and UTP. One possible explanation for this enhanced incorporation is as follows: in the evolutionary experiment without dNTPs, the RNAPs did not have to distinguish between canonical nucleotides, NTPs, and non-canonical nucleotides, dNTPs. Therefore, mutations that impaired the ability to discriminate between NTPs and dNTPs could have accumulated. Such mutations might distort the structure or electric distribution around the nucleotide binding pocket, one of the selection mechanisms for NTPs [22]. In addition, it has been reported that incorporation fidelity can be changed by mutations in residues that do not directly interact with nucleotides [23]. The mutations introduced into the RNAPs used here might distort the structure and allow the accommodation of noncanonical nucleotides, including EdUTP and EUTP.

During the early stages of life before the advent of DNA, RNAP might have evolved in the absence of dNTPs and thus might not have had strict substrate specificity. Such loose substrate specificity might have been a common feature of ancient enzymes [12]. If the hypothesis is correct, the transition of genetic material from RNA to DNA might have been easier than previously thought because such RNAP could have started synthesizing DNA soon after dNTPs were supplied by one process or another, such as the reverse of the deoxyriboaldolase reaction [24]. Although in our experiment, the dNTP incorporation ratio was still low (approximately 0.3% at most), the results of this study provide new insights into the feasibility of the RNA-to-DNA transition.

The transition from RNA replication to DNA replication also requires changes in template specificity; RNAP must have used DNA instead of RNA as a template. Thus far, we have not obtained any results showing that mutant RNAPs can use DNA as a template. Understanding changes in template specificity is an important challenge, and we need further evolutionary experiments to elucidate this matter.

## 4. Materials and Methods

### 4.1. RNA Preparation

Plasmids encoding the RNAs used in this study were constructed as described previously [17,18]. DNA encoding the T7 promoter and β subunit of Qβ replicase was amplified by PCR and digested with SmaI (Takara, Japan). RNA was synthesized from the DNA by in vitro transcription with T7 RNA polymerase (Takara) and purified using the RNeasy Mini Kit (QIAGEN). The original RNA, the same as the initial RNA in a previous study [17], was synthesized from the plasmid pUC-mdv(-)beta(+). RNA1, the same as the dominant RNA at round 128 in a previous study [17], was synthesized from the plasmid pUC-N96(+). RNA2 and RNA3, the same as the Host-99 and Host-115 RNAs in a previous study [18], were synthesized from the plasmids pEX-Host99 and pEX-Host115, respectively.

### 4.2. RNA Replication and Measurement

Each RNA (10 nM) was incubated at 37 °C for 2.5 h in the reconstituted translation system of *E. coli* [14] prepared according to a previous study [17], except for UTP concentration (1.25 mM UTP in Figure 1c, 1.1 mM EdUTP, and 0.11 mM UTP in Figure 2b,c, and 1.1 mM EUTP and 0.11 mM UTP in Figure 2d,e and Appendix A). The mixture was then diluted 10,000-fold with 1 mM EDTA (pH 8.0), and RNA concentration was measured by RT-qPCR with a One Step TB Green PrimeScript RT-PCR Kit (Takara, Japan) using the following primers: 5′-GCTGCCTAAACAGCTGCAAC-3′ and 5′-CGCTCTTGGTCCCTTGTATG-3′.

### 4.3. Labeled RNA Recovery

The mixture (18 µL) after the replication reaction was purified using the RNeasy Mini Kit (QIAGEN) and eluted with 21 µL water. The ethynyl group on the RNA was conjugated with biotin by click reaction using a Click-it Nascent RNA Capture Kit (Invitrogen). The mixture was rotated gently around 23 °C for 30 min and then purified in the same manner as described above. The elution was carried out with 27 µL water. The purified RNA was mixed with 10 µL of streptavidin-coated magnetic beads (Dynabeads MyOne Streptavidin T1) and rotated gently at room temperature for 30 min. The beads were then washed 10 times using a magnet on ice. The beads were suspended in 10 µL of phosphate-buffered saline containing 0.1% sodium dodecyl sulfate solution and heated at 95 °C for 5 min to separate the RNA from the beads. After recovery, the RNA solution was diluted 100-fold with 1 mM EDTA aqueous solution and analyzed by RT-qPCR as described above.

## Figures and Tables

**Figure 1 life-12-00032-f001:**
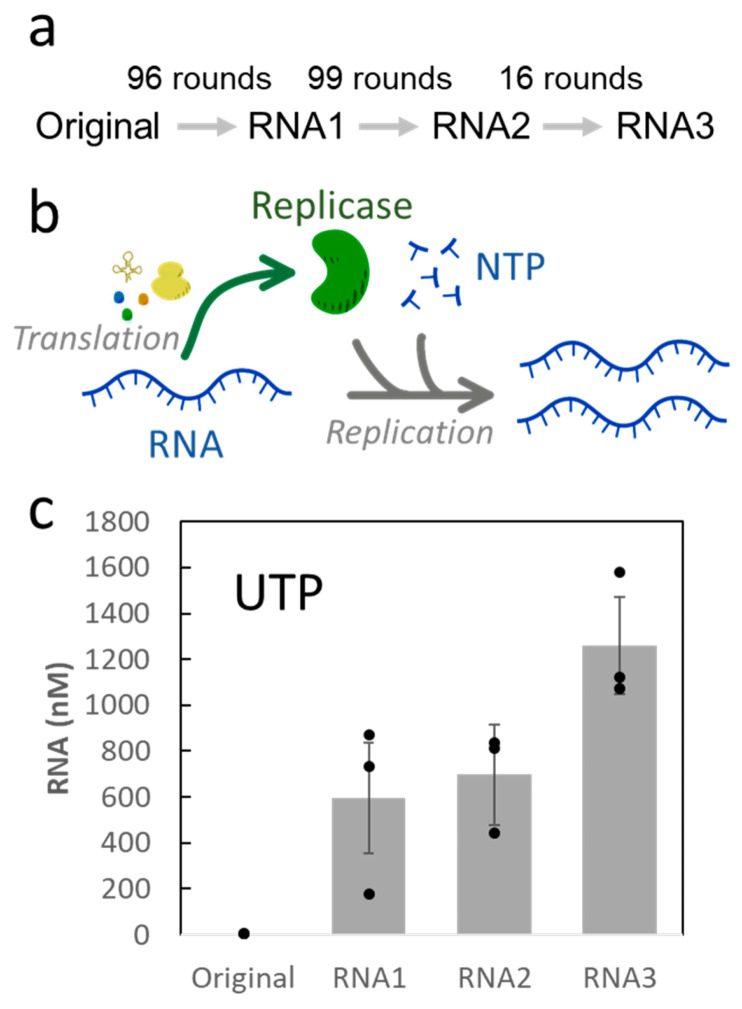
(**a**) Relationships of the RNAs used in this study. These are the dominant RNAs that appeared in the course of the previous serial dilution experiments (Original and RNA1 were obtained from [17], and RNA 2 and RNA3 were obtained from [18]). (**b**) Scheme of RNA replication. The Qβ replicase is translated from the RNA in a reconstituted translation system of *Escherichia coli* and then replicates the RNA. The step for complementary strand synthesis is omitted for simplicity. (**c**) RNA concentration after replication. Each RNA (10 nM) was incubated at 37 °C for 2.5 h in the reconstituted translation system, including 1.25 mM UTP. The error bars represent standard error (*n* = 3).

**Figure 2 life-12-00032-f002:**
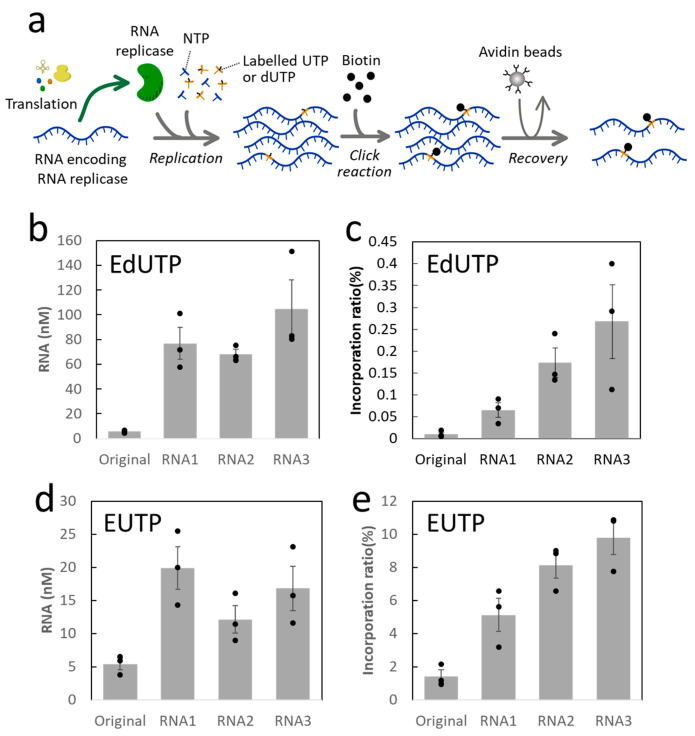
RNA replication and the incorporation ratios in the presence of EdUTP and EUTP. (**a**) Experimental procedure for the detection of EdUTP or EUTP incorporation. First, each RNA (10 nM) is replicated by the RNA polymerase translated from itself in the presence of EdUTP or EUTP. Second, the ethynyl-labeled RNA is conjugated with biotin by click reaction. Third, the biotinylated RNA is recovered using avidin-coated magnetic beads. The RNA concentrations before and after the recovery step are measured by RT-qPCR. (**b**) RNA concentrations before the recovery with 1.1 mM EdUTP and 0.11 mM UTP. (**c**) EdUTP incorporation ratios in the replicated RNAs, i.e., the ratios of the RNA concentrations after the recovery to those before the recovery, of the RNAs replicated in b. (**d**) RNA concentrations before the recovery with 1.1 mM EUTP and 0.11 mM UTP. (**e**) EUTP incorporation ratios, i.e., the ratios of the RNA concentrations after the recovery to those before the recovery, of the RNAs replicated in d. The error bars represent standard error (*n* = 3).

**Figure 3 life-12-00032-f003:**
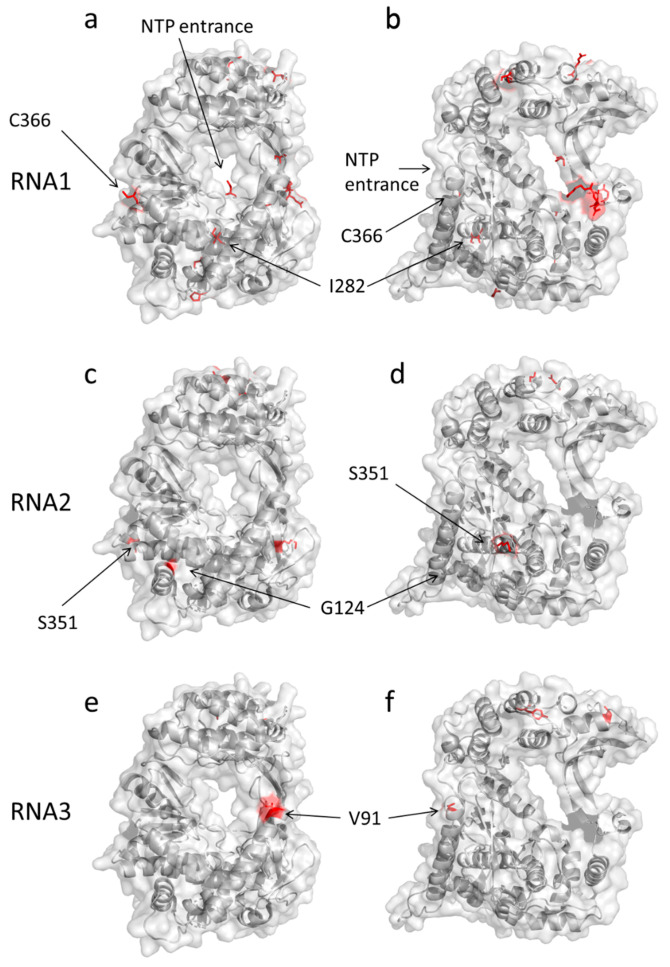
Location of the fixed mutations in each RNA polymerase. Front (**a**,**c**,**e**) and left side (**b**,**d**,**f**) views of the β subunit of Qβ replicase showing the location of the newly fixed mutations in (**a**,**b**) RNA1, (**c**,**d**) RNA2, and (**e**,**f**) RNA3 in red. The 3D structure is taken from Protein Data Bank 3AGP.

## Data Availability

Data is contained within the article or Appendix A.

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
