# Peer review of "Relaxed Substrate Specificity in Qβ Replicase through Long-Term In Vitro Evolution"

_life, 2021, doi:10.3390/life12010032_

Round 1
Reviewer 1 Report
The manuscript by Yukawa et al. describes the increased ability of an evolved/mutated version of the RNA-dependent RNAP Qβ to incorporate non-canonical nucletiode triphosphates. The authors describe the mutations at multiple sites that is found in such "evolved" RNAPs. It is somewhat disappointing that none of the identified mutants appears to make a substantial structural/functional difference on its own but only work in combination.
In summary, the manuscript describes an interesting experimental set-up and promising results that are publishable as a short report. I believe, however, that the presentation of the manuscript can be clarified to make the content more accessible to readers.
Suggested changes/improvements:
- I only fully started to understand the strategy behind the paper when reading the first two paragraphs of the Discussion towards the end of the paper. I suggest that some of these more detailed explanations should be provided right at the beginning. When I read the title and abstract, I assumed that the authors select for increased dNTP incorporation due to the presence of excess dNTPs (which is how positive selection would normally work) and then I only realized later that the authors removed selection pressure by keeping dNTPs completely out of the reaction - a somewhat unusual (but original!) way of doing such an experiment ...
- Results: I do not fully understand how the repression of the parasitic RNA works and what effect it has on the evolution. The authors should spell this out in more detail and explain what they assume happens during the switch in the method between amplification cycles between 99 and 115 rounds. Also, maybe the authors could comment why they use the repetitive amplification method for the evolution, rather than in vitro mutagenesis - are there specific reasons for such a choice? Finally, some of my confusion arose from the use of the term "in vitro evolution" in line 58, which implies (in my mind) specific selection towards a particular desired phenotype. As far as I understand, the authors are not really "evolving" the system but are using it to generate mutant variants.
- Also, to aid clarity I suggest that the authors include a schematic diagram of the amplification method (similar to Fig. 2a) as the first panel in Fig. 1 so that the process becomes clearer.
- Line 87 "Original RNA" should be written "original RNA" (no capitals)
- I do not understand why the authors choose ethynyl-labelled dNTP to measure incorporation of dNTPs. Would it not be much easier to assay dNTP incorporation using e.g. 32P-labeled dNTPs that are chemically completely identical to dNTPs? Can they provide a diagram of the structure of the ethynyl nucleotide variants and how the click-chemistry works?
- line 141: this needs to be written more clearly ("reverse mutants of each of the mutations" sounds a bit strange). I assume that the authors are saying that they reverted the mutated residues back to the wildtype identiy?
Reviewer 2 Report
In this paper the authors discovered that bacteriophage RNA polymerases (RNAPs) that had been subjected to in vitro evolution in an artificial RNA replication system became able to incorporate ethynyl-labeled dUTP at significant levels. Testing an original RNA and its three mutants from increasing rounds of evolution, they showed a positive correlation between the rounds of evolution and the capabilities of dUTP incorporation. The authors interpreted that RNAPs reduced the substrate specificity during in vitro evolution that proceeded in the absence of dNTP. They used experimentally obtained data to suggest that the genetic material could be changed from RNA to DNA in a relatively easy way. I feel that despite many assumptions in their story, the authors’ efforts and findings in this work are interesting and of scientific value, contributing to our understanding of how a DNA-based living system occurred. The manuscript is concisely and smartly written, but I have several concerns that should be addressed by the authors before publication.
1) The title is a little misleading. To my understanding, the main conclusion of this work is that RNAPs reduce substrate specificity if they have experienced in vitro evolution in the absence of dNTPs. The present title, however, might give an impression that RNAPs acquire an ability to incorporate dUTP if they evolve in the presence of dUTP. In relation to this, what happens if RNAPs evolve in the presence dNTPs or dUTP in the in vitro system.
2) Fig2b, d: Why was the absolute mount of RNA obtained with EUTP much lower than that obtained with EdUTP? To provide a possible explanation for this may help readers to fully understand the significance of the experiments. In relation to this, I cannot follow the statement “.., similar to the results obtained without using EdUTP (Fig.1b).” (line 102)
3) In the manuscript, including the title and abstract, the authors use the word “tend” several times but it makes the statements ambiguous depending on the context. I speculate that the authors want to stress the results from increasing rounds of in vitro evolution. However, when the experimental design is not provided, the statements including “tend” are hard to understand. In addition, the authors tested only three selected mutant RNAs from the in vitro evolution experiment. I am not sure whether the difference between RNA2 and RNA3 in the EdUTP incorporation ratio (Fig. 2c) is significant.
4) dUTP is not a component of natural DNA. dNTP usually indicates dATP, dTTP, dCTP, and dGTP. A more careful interpretation is required.
Minor points
5) Fig2a: RNA encoding RNA -> RNA encoding RNAP
6) Fig1, 2 bar graphs: as N is small, the individual values should be additionally plotted.
7) Supplementary table S1 and Fig S1, S2: Which is correct, I282 or I283?
